# A Genome-Wide Alternative Splicing Landscape Specifically Associated with Durable Rice Blast Resistance

Dong Liang, Junjie Yu, Tianqiao Song, Rongsheng Zhang, Yan Du, Mina Yu, Huijuan Cao, Xiayan Pan, Junqing Qiao, Youzhou Liu, Zhongqiang Qi *,† and Yongfeng Liu *,†

Jiangsu Academy of Agricultural Sciences (JAAS), Nanjing 210014, China
* Correspondence: 20130019@jaas.ac.cn (Z.Q.); 19970013@jaas.ac.cn (Y.L.)
† These authors contributed equally to this work.

**Abstract:** The rice blast, caused by the hemibiotrophic plant pathogen *Magnaporthe oryzae*, is a devastating disease that threatens rice crop production worldwide. The molecular interactions that underlie the rice-*M. oryzae* interaction have received much attention. However, genome-wide research focusing on alternative splicing (AS) has not been well-studied in rice—*M. oryzae* interactions. AS in plants leads to diverse proteomes without an expansion in gene numbers to regulate cellular processes during abiotic or biotic stress. The *Pi21* gene negatively regulates rice resistance to *M. oryzae* infection, and thus the *Pi21*-RNAi silenced transgenic line (#241) exhibits partial but durable resistance. We compared the AS landscape in #241 and "Nipponbare" (Nip) during interacting with *M. oryzae* Guy11, and the alternative 3′ splice-site (A3SS) is the most common AS type. GO enrichment analysis of #241-specific differentially alternatively spliced genes (DASGs) revealed that WRKY transcription factors (TFs), bHLH TFs, F-box protein with leucine rich repeats, AAA-type ATPase, and protein kinase were enriched in the GO terms "response to jasmonate acid (JA)" and "ethylene (ET)" at 24 h post-inoculation (hpi). At 48 hpi, one #241-specific DASG, ubiquitin gene (*Os08g0295000*), was predicted to be involved in endoplasmic reticulum (ER) stress. In silico analysis combined with PCR amplification confirmed that multiple isoforms are produced by *Os08g0295000* and a skipped exon (SE) event results in isoform switching during interaction with *M. oryzae*. Protein–protein interaction (PPI) network analysis predicted that *Os08g0295000*-encoding proteins may interact with SNARE protein Q9LGF8 (Uniprot ID) to cooperatively regulate rice's response to *M. oryzae*. This study uncovered the AS profile of rice in response to *M. oryzae*, which will help to explore the linkage between AS and durable rice resistance.

**Keywords:** interaction; alternative splicing (AS) landscape; rice durable resistance; differentially alternatively spliced genes (DASGs)

## 1. Introduction

Alternative splicing (AS) describes the mechanism by which multiple transcript isoforms are produced from single precursor mRNAs (pre-mRNAs) [1], and results mainly from the different combinations of exons for inclusion into the mRNAs. AS increases the proteome diversity without massive gene gain events [2]. The translation of alternatively spliced transcripts determines which domains are present in the corresponding proteins, and this can result in a diverse higher structure of proteins with different functions [3]. The spliceosome is a ribonucleoprotein complex consisting of different RNP subunits that can accurately recognize exons and introns to carry out the splicing reaction, resulting in AS [4]. Genome-wide studies have shown that AS occurs widely in eukaryotes. More than 95% of human multi-exon genes undergo AS [5]. In *Drosophila melanogaster*, alternative splicing results in tissue- or growth stage-specific protein isoforms with different functions in development [6]. Based on high-throughput sequencing, 61% of intron-containing *Arabidopsis* genes undergo AS events under multiple conditions [7], such as development and abiotic

and biotic stress. Previous research report that several mineral nutrition-related genes in rice [8], such as the sulfate transporter gene *Os03g06520*. *Os03g06520* undergoes AS in response to mineral deficiencies. Furthermore, it was reported that AS plays a role in the *Arabidopsis thaliana* defense mechanism against infection by *Pseudomonas syringae* [9,10].

The rice blast pathogen *Magnaporthe oryzae* is the cause of a destructive disease that can reduce yields by 30% wherever rice is grown worldwide [11,12]. During the past several decades, the zig-zag model was proposed to describe plant–pathogen interactions; there are two types of plant immune systems [13]: (1) pathogen-associated molecular pattern (PAMP)-triggered immunity (PTI) and (2) effector-triggered immunity (ETI) (the plant immune system). The plant's response to pathogens is a complex system that consists of multiple gene families. Approximately 25 rice resistance (*R*) genes that mainly encode nucleotide binding site-leucine-rich repeat (NBS-LRR) proteins were found to recognize effectors produced by pathogens. In addition, 77 rice defense-regulator (DR) genes are also required for pathogen perception, signaling transduction, and other downstream responses to stimulation of hormones (salicylic acid (SA), jasmonate acid (JA), and ethylene (ET)), the hypersensitive response (HR), and reactive oxygen species (ROS) accumulation [14]. These DR genes include pattern-recognition receptors (PRRs), chitin oligosaccharide sensing factors, cell wall-associated kinases, MAPK cascades, and WRKY transcription factors (TFs). AS was found to occur in several of these resistance-associated genes in previous research. Two rice WRKY TF genes, *OsWRKY62* and *OsWRKY76*, underwent AS events in response to *M. oryzae* and *Xanthomonas oryzae* pv *oryzae*, which reduced sequences of the W box motif and repressed canonical spliced variants [15]. AS events in 119 rice NBS-LRR genes were identified by BLAST searches against genome sequences [16]. However, genome-wide identification of AS events in defense-associated genes still remains unexplored by high-throughput sequencing data, which provides more reliable AS detection than the use of in silico methods alone.

In contrast to race-specific resistance, partial but durable rice resistance was contributed to by quantitative trait loci (QTLs) [17]. Among these beneficial QTLs, *Pi21* was cloned by [18]. According to [18], the *Pi21* gene encodes a cytoplasmic proline-rich protein with a putative metal-binding domain. The wild-type *Pi21* gene inhibits the rice defense response, and loss of the proline-rich motif of *Pi21* can promote the plant defense response. This suggests that the susceptible *Pi21* allele negatively regulates rice disease resistance and represents a susceptibility factor. The durable non-race-specific rice blast resistance of the *Pi21*-RNAi transgenic line (#241, background cultivar is "Nipponbare") was verified by [19]. In addition, the conserved mechanism of the "Nipponbare" (Nip) response to four different *M. oryzae* strains was proposed in our previous publication [20].

In this study, public transcriptome data for #241 and the Nip control were collected. Then, a transcriptome assembly pipeline was performed for the identification of AS events associated with durable rice blast resistance, which includes annotated genes, novel genes, or isoforms being absent in the reference genome. By comparing the AS landscape of #241 and Nip, we identified 187 and 126 differentially alternatively spliced genes (DASGs) specific to #241 at 24 and 48 h after inoculation (hpi) with *M. oryzae*, respectively. These #241-specific DASGs are mainly involved in response to plant hormone stimulation, endoplasmic reticulum stress, and encode protein kinases, LRR-containing proteins, WRKY TFs, bHLH TFs, ubiquitin, and thioredoxin. Among them, the ubiquitin coding gene *Os08g0295100* was confirmed to have undergone a skipped exon (SE) of AS event in response to *M. oryzae*.

## 2. Results

### 2.1. Identification of Rice Differentially Alternatively Spliced Genes (DASGs) during Interaction with Magnaporthe oryzae

To investigate transcriptional changes and the AS scheme of defense-associated genes, we first collected the publicly-available transcriptomes of the *Pi21*-RNAi transgenic rice line (#241) and the susceptible cultivar "Nipponbare" that had been infected by *M. oryzae* Guy11 (SRA492222) [19]. It has been suggested that *Pi21* negatively regulates rice blast

resistance, and plants carrying a loss-of-function mutation of *Pi21* express durable and non-race-specific rice blast resistance [18]. In this study, the transcriptome data used for AS detection included the following samples: (i) #241 infected by *M. oryzae* Guy11 at 24 and 48 hpi (sample names: Guy11_241_24 h and Guy11_241_48 h); (ii) Nip infected by the *M. oryzae* strains Guy11 at 24 and 48 hpi (sample names: Guy11-Nip-24 h, Guy11-Nip-48 h); (iii) the un-infected samples of #241 and Nip were defined as the controls (sample names: Guy11-241-0 h and Guy11-Nip-0 h). Due to the pathogen perception by rice plants that occurs within 48 h [21], selection of the 24 and 48 hpi timepoints will help us to uncover links between AS and activation/inhibition of rice innate immunity.

We designed a transcriptome assembly pipeline to detect AS events (Figure 1a). Of the trimmed clean reads, 91.58% to 93.93% mapped to the rice reference genome with HiSAT2 (step II) (Table 1). Using the mapped reads, transcripts were assembled with StringTie. A total of 68,103 transcripts (across 39,231 genes) were yielded and 59,026 of them (across 33,902 genes) had coding potential, which were retained for the next step (step III, IV). Using gffcompare, retained assembled transcripts were compared with annotated transcripts, which produced comparison class codes for distinguishing each other following the method proposed by [22]. In total, we identified 39,224 annotated transcripts (from 32,562 reference genes), 18,650 novel isoforms (from 9929 reference genes), and 1054 novel isoforms (from 910 novel genes) (Figure 1b). Based on the expression level of the obtained transcripts/isoforms and the corresponding genes, we identified 7802 differentially expressed genes (DEGs) across all samples by DEGseq (Figure 1a) (step V), and 2504 of them were also predicted as alternatively spliced genes (ASGs) by SUPPA2 [23], which were defined as differentially alternatively spliced genes (DASGs).

### 2.2. Comparative Analysis Uncovered More DASGs in Pi21-RNAi Transgenic Line (#241)

Through comparison of Guy11_241_24 h vs. 0 h and Guy11_241_48 h vs. 0 h, there were 397 and 364 DASGs in the Guy11_241_24 h vs. 0 h and Guy11_241_48 h vs. 0 h, respectively (Figure 1c), which were more than that of the Nip—*M. oryzae* interaction (329–336 DASGs). In the Guy11_241_24 h vs. 0 h comparison, there were 397 DASGs (301 up-regulated and 96 down-regulated) and 364 DASGs (294 up-regulated and 70 down-regulated) in the Guy11_241_48 h vs. 0 h comparison. Meanwhile, 336 DASGs (245 up-regulated and 91 down-regulated) were identified in the Guy11_Nip_24 h vs. 0 h comparison and 329 DASGs (262 up-regulated and 67 down-regulated) in the Guy11_Nip_48 h vs. 0 h comparison. Furthermore, the total number of AS events in #241 was 493 (24 hpi) and 441 (48 hpi), although there were only 408 (24 hpi) and 392 (48 hpi) in Nip (Figure 1d). SUPPA2 classified the AS events into the following seven types: (1) skipped exon (SE), (2) mutually exclusive exons (MX), (3) alternative 5′ splice-site (A5SS), (4) alternative 3′ splice-site (A3SS), (5) retained intron (RI), (6) alternative first exon (AF), and (7) alternative last exon (AL). As shown in Figure 1d, alternative donor (A3SS) was the dominant AS type in all samples, followed by retained intron (IR), alternative acceptor (A5SS), and skipped exon (SE).

To identify #241-specific DASGs, we compared DASGs identified in #241 and Nip. As Figure 2 showed, 226 DASGs were specific to #241 at 24 hpi (Table S2) and 215 at 48 hpi (Table S3). The AS events that occurred in the #241-specific DASGs may involve general response mechanisms that are specific to resistible cultivars. Thus, these #241-specific DASGs were absent in Nip and were retained for further analysis to explore the linkage between AS events and durable rice blast resistance.

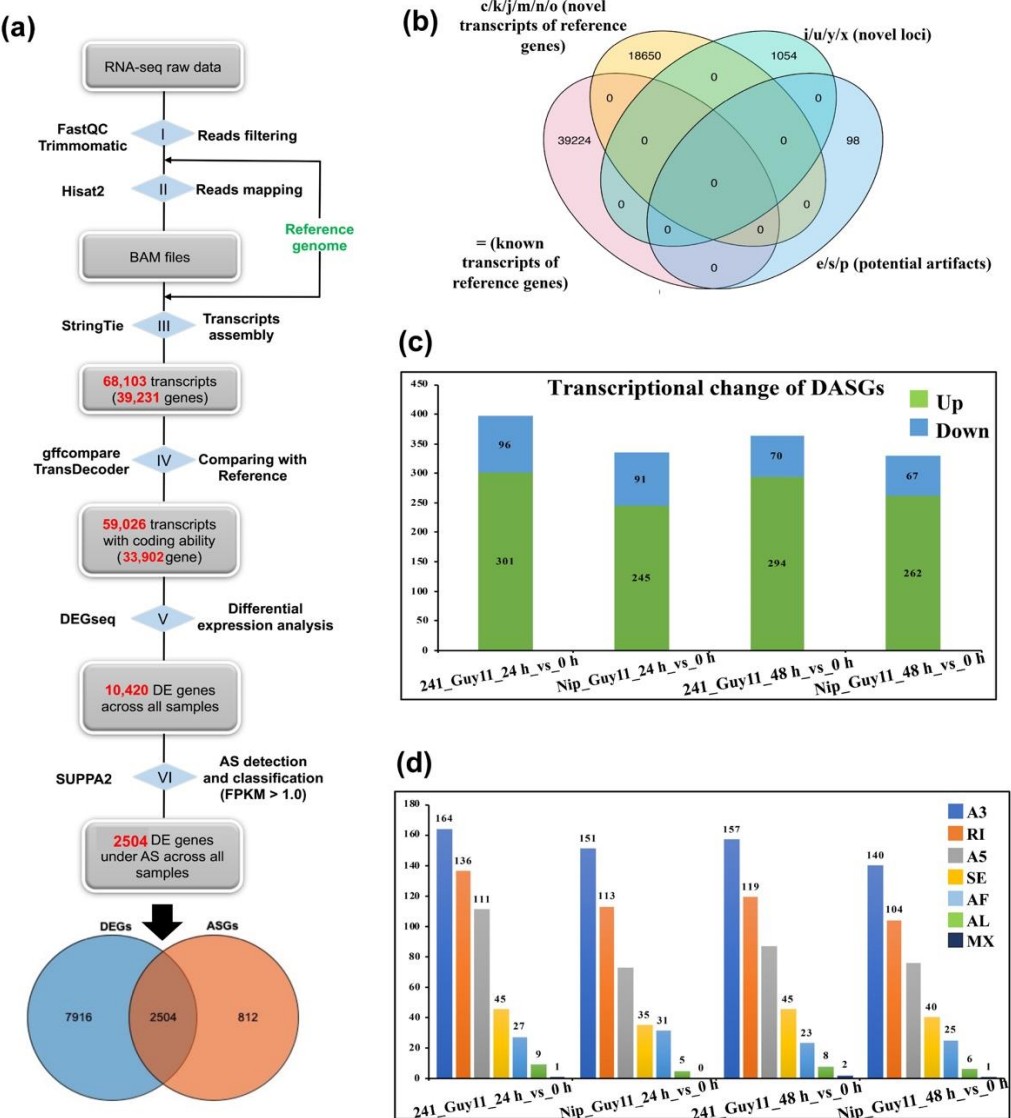

**Figure 1.** Identification and distribution of alternatively spliced genes (ASGs) in rice. (**a**) Computational pipeline for genome-wide detection of alternative splicing (AS) events and differentially alternatively spliced genes (DASGs). (**b**) Venn diagram of comparison of StringTie-produced transcripts and annotated transcripts by gffcompare Kit. (**c**) Distribution of differentially expressed genes (DEGs) across all samples. (**d**) The numbers of all seven types of alternative splicing events across all samples. A3: Alternative 3′ splice-site; RI: Retained intron; A5: Alternative 5′ splice-site; SE: Skipped exon; AF: Alternative first exon; AL: Alternative last exon; MX: Mutually exclusive exons.

**Table 1.** Data accession and mapping rate of data used in this study.

| Interaction Samples | Run Accession | Data Type | Raw Reads | Trimmed CLEAN Reads | Mapped Reads | Overall Mapping Rate |
|---|---|---|---|---|---|---|
| Guy11-241-0 h * | SRR5007209 | single-end | 10,642,019 | 9,897,077 | 9,219,559 | 93.15% |
| Guy11-241-24 h * | SRR5007145 | single-end | 10,666,277 | 9,652,980 | 9,010,345 | 93.34% |
| Guy11-241-48 h * | SRR5007152 | single-end | 11,016,907 | 9,945,723 | 9,316,447 | 93.67% |
| Guy11-Nip-0 h * | SRR5007266 | single-end | 11,114,533 | 10,614,225 | 9,818,047 | 92.50% |
| Guy11-Nip-24 h * | SRR5007321 | single-end | 10,502,187 | 9,872,055 | 9,040,986 | 91.58% |
| Guy11-Nip-48 h * | SRR5007327 | single-end | 11,369,315 | 10,459,769 | 9,825,296 | 93.93% |

* *M. oryzae* strains-rice cultivar-time points.

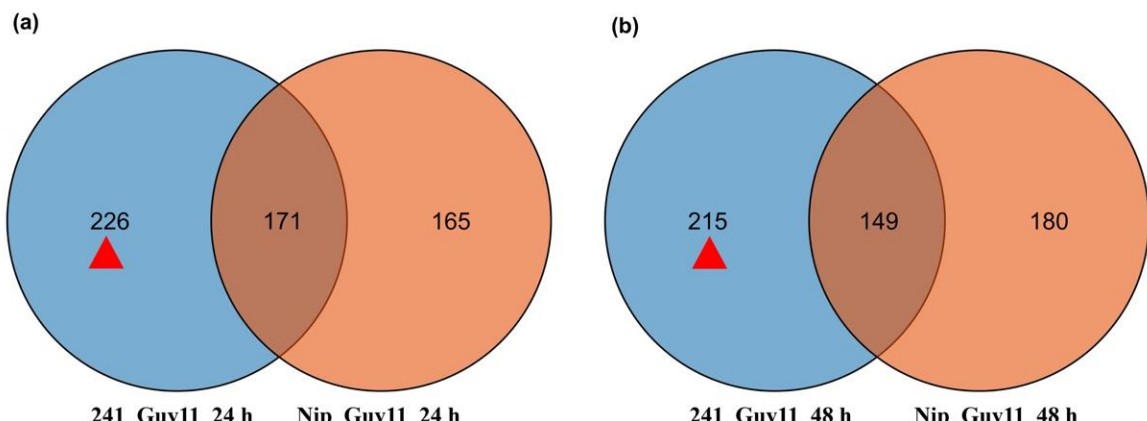

**Figure 2.** Comparison of differentially alternatively spliced genes (DASGs) of "Nipponbare" (Nip) and *Pi21*-RNAi transgenic line during interaction with *M. oryzae* Guy11 (24 hpi and 48 hpi). (**a**) At 24 hpi. (**b**) At 48 hpi. The red triangles are the #241-specific DASGs. Red triangles represent numbers of #241-specific DASGs at 24 and 48 hpi.

### 2.3. #241-Specific DASGs Mainly Associated with Plant Hormone Stimulation Response and Endoplasmic Reticulum Stress

We performed gene ontology (GO) enrichment analysis of #241-specific DASGs to explore their putative functions. As Figure 3a and Table S4 showed, we found that seven resistance-associated GO terms were enriched by #241-specific DASGs at 24 hpi. These GO terms mainly include responses stimulated by plant hormones, such as jasmonic acid (JA) and ethylene (ET), which are essential molecules for the induction of host plant non-specific resistance to plant pathogens [24]. Eight up-regulated and four down-regulated #241-specific DASGs were found to be enriched in these GO terms (Figure 3a), which mainly encode protein kinase (Pkinase: PF00069), leucine-rich repeat containing proteins (LRR_6: PF13516), ATPases (AAA_12: PF13087), WRKY TFs (WRKY: PF03106), and bHLH TFs (HLH: PF00010). At 48 hpi, three up-regulated #241-specific DASGs were enriched in GO terms of "response to endoplasmic reticulum stress", which included DASGs encoding ubiquitin protein (Ubiquitin: PF00240) and thioredoxin protein (Thioredoxin: PF00085). Four up-regulated #241-specific DASGs, mainly encoding RNP-1 (RRM_1: PF00076) and DNA mismatch repair protein (Mlh1_C: PF16413), were enriched in "immune system development" (Figure 3b).

*Os08g0295100*, the ubiquitin protein coding gene, was up-regulated at 48 hpi of the #241-*M. oryzae* interaction and might be associated with endoplasmic reticulum stress (Figure 3b). One annotated transcript (*Os08t0295100-00*) and four novel isoforms (*MSTRG.25458.1, MSTRG.25458.2, MSTRG.25458.3,* and *MSTRG.25458.6*) were produced by Os08g0295100 (Table S3). The AS profile of *Os08g0295100* discovered that a skipped exon (SE) alternative splicing event occurred in Chr8:11871192-11872220 and Chr8:11872573-11873297, which was marked in a Sashimi plot (Figure 4a). Transcript-level expression analysis revealed that the highest expressed transcript/isoform switched from *MSTRG.25458.2* to *MSTRG.25458.1* at the onset of infection by *M. oryzae* (Figure 4c). PCR amplification confirmed the existence of *MSTRG.25458.2, MSTRG.25458.1,* and *MSTRG.25458.3* (Figure 4d). Primers used for PCR amplification are provided in Figure S1, which reveals that two skipped-exon regions (L1 and L2 fragments) result in a different coding sequence (CDS) length: *MSTRG.25458.2* (1020 bp), *MSTRG.25458.1* (630 bp) and *MSTRG.25458.3* (984 bp). Furthermore, interacting proteins of Os08g0295100 were predicted through protein–protein interaction (PPI) network analysis based on the STRING database [25]. A ubiquitin-like protein (A0A0P0X6D6) and SNARE/syntaxin-5 protein (Q9LGF8) was predicted to interact with Os08g0295100 (Figure 4b).

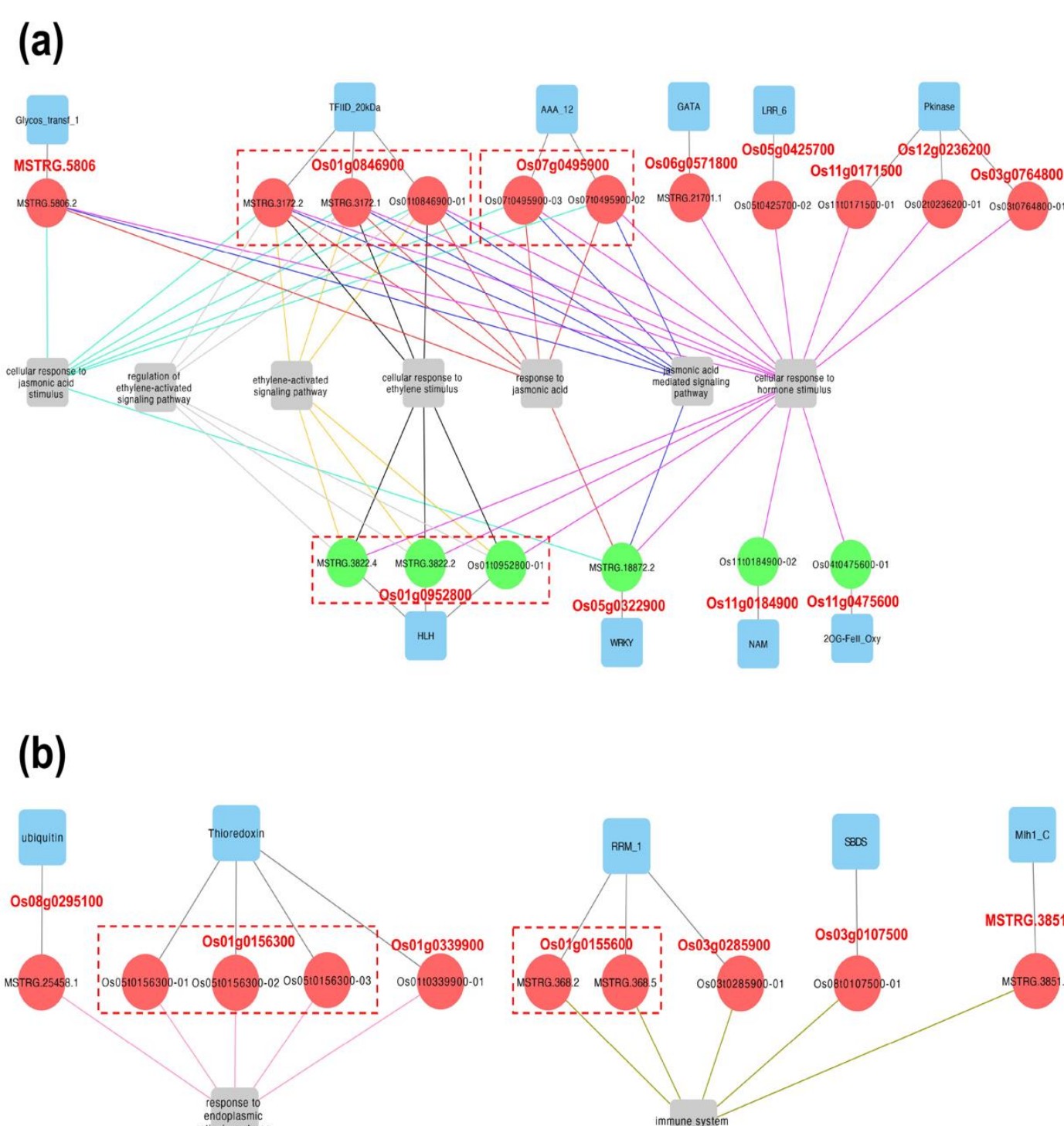

**Figure 3.** Defense-associated enriched Gene Ontology (GO) terms of #241-specific DASGs with adjusted *p*-value < 0.05 and GO level > 5 at (**a**) 24 hpi and (**b**) 48 hpi. Red dashed boxes marked alternative transcripts/isoforms produced by the same #241-specific DASGs, of which their accessions were listed. Circles with a red color mean up-regulated transcripts/isoforms; circles with a green color mean down-regulated transcripts/isoforms.

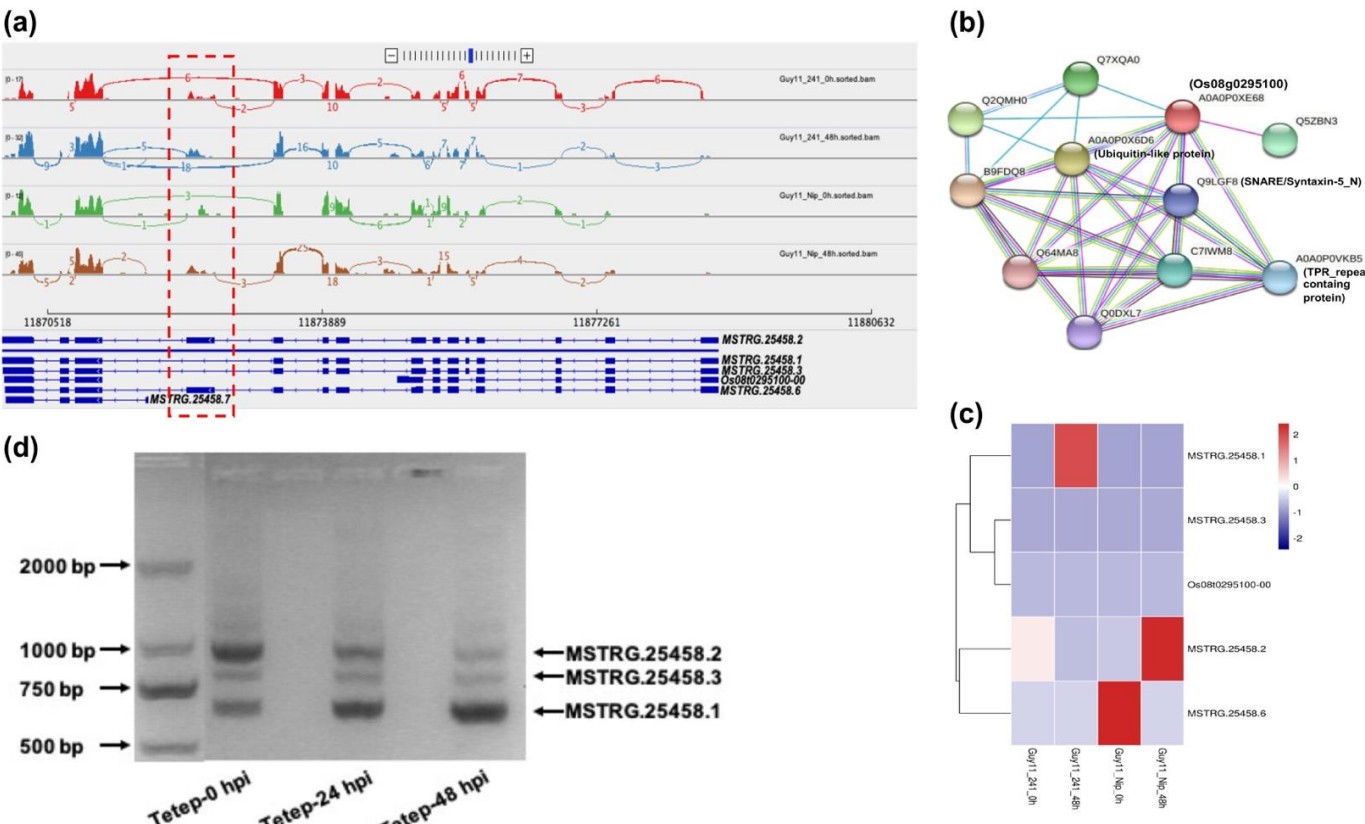

**Figure 4.** Sashimi plots and protein–protein network (PPI) analysis for Os08g0295100. (**a**) Sashimi diagram showing the exon-intron structure of representative annotated transcripts and novel isoforms. The red dashed box encloses the location of the AS events. (**b**) The PPI network for the protein produced by *Os08g0295100*. Green, red, and blue lines indicate interactions predicted from gene neighborhood, fusions, and co-occurrence. Light green, black, and dark blue lines indicate additional interactions inferred from text mining, co-expression, and protein homology. (**c**) Expression profile of an annotated transcript and novel isoforms produced by *Os08g0295100*. (**d**) PCR amplification of transcripts/isoforms generated by *Os08g0295100* at 0, 24, and 48 hpi.

## 2.4. Investigation of Infection-Specific AS Isoforms Generated by #241-Specific DASGs

During the analysis of the expression level of #241-specific DASGs-generated transcripts/isoforms at each stage, we introduced the concept of the transcript usage index (TUI), which indicates the expression level of individual transcripts compared to the corresponding gene (TUI = FPKMtranscript/FPKMgene) [26]. The #241-specific DASGs-generated transcripts/isoforms were divided into two groups based on their TUI value: (1) primary transcripts/isoforms ($0.5 \leq$ TUI $< 1$); (2) secondary transcripts/isoforms ($0 <$ TUI $< 0.5$). As Figure 5a showed, we found 36 primary transcripts/isoforms (from 36 DASGs) at 24 hpi and 49 (from 49 DASGs) at 48 hpi during the response to *M. oryzae* (Table S5), which were also marked by red dotted lines in Figure 5a,b. The average TUI value of these primary transcripts/isoforms was 0.75 at 24 hpi and 0.78 at 48 hpi (Figure 5c,d). Through domain annotation of primary transcripts/isoforms mentioned above based on the PFAM database (https://pfam.xfam.org/, accessed on 6 April 2022), the #241-specific primary transcripts/isoforms at 24 hpi are predicted to encode development/cell death associated proteins (Dev_Cell_Death: PF10539), PPR repeat containing proteins (PPR: PF01535), Leucine-rich repeat proteins (LRR_6: PF13516), kinase (Pkinase: PF00069), F-box domain containing proteins (F-box: PF12937), zinc finger TFs (ZF-CCCH: PF00642) and ubiquitin protein (ubiquitin: PF00240) (Table 1). At 48 hpi, #241-specific primary transcripts/isoforms are predicted to mainly encode development/cell death-associated proteins (Dev_Cell_Death: PF10539), leucine-rich repeat containing proteins

(LRR_6: PF13516), ubiquitin protein (ubiquitin: PF00240), TFs of zinc finger (ZF-C2H2: PF00096), zinc finger transcription factors (zf-C2H2: PF00096), and MYB (Myb_DNA-binding: PF00249).

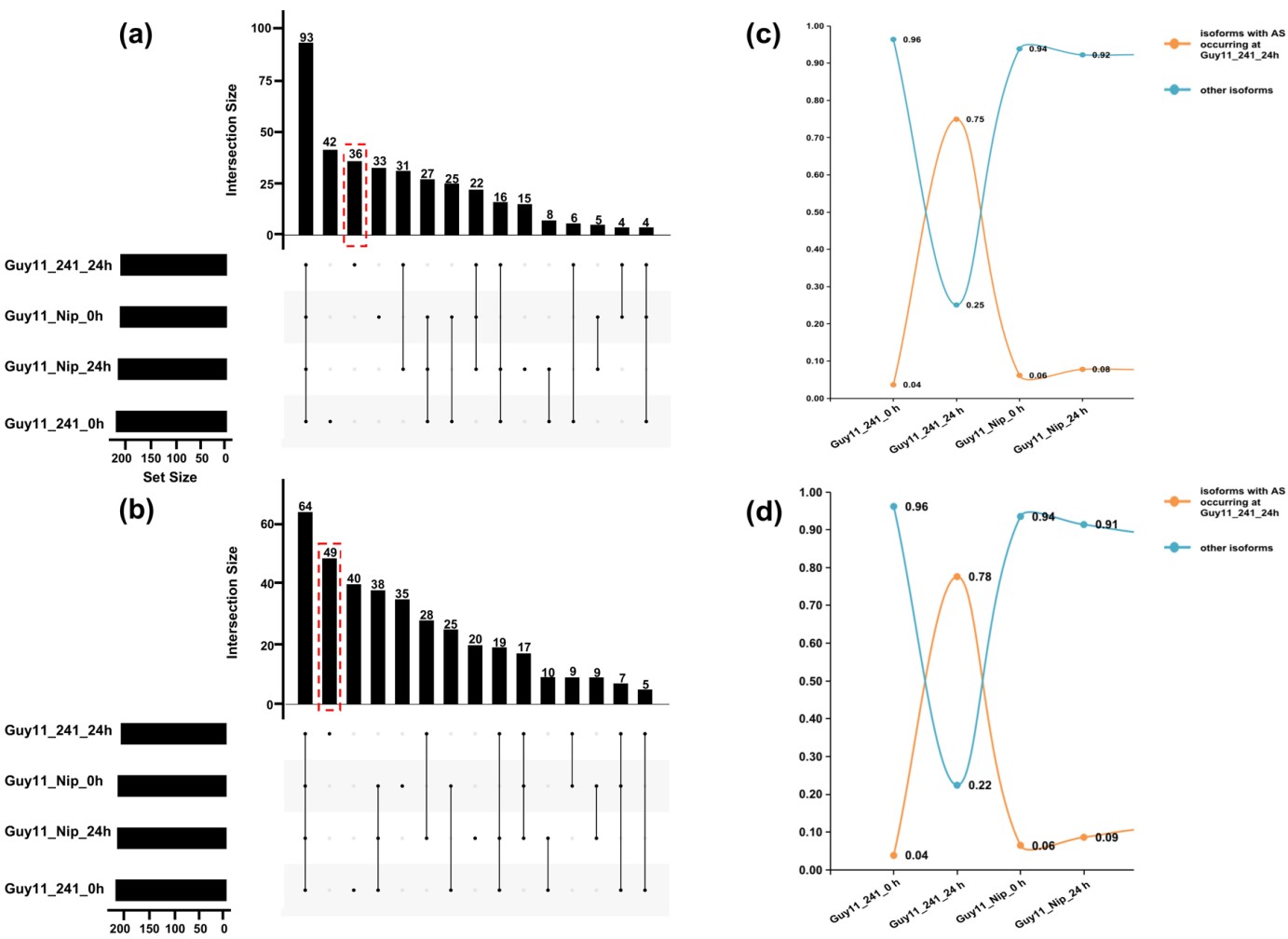

**Figure 5.** Distribution and average TUI values of primary transcripts/isoforms across all samples (**a**,**b**). Upset Venn plots showing overlapping primary annotated transcripts or novel isoforms at 24 hpi (**a**) and 48 hpi (**b**) in #241 and Nip. Red dashed box encloses infection-specific primary transcripts/isoforms in #241. (**c**,**d**) Average transcript usage index values for #241-specific primary annotated transcripts or novel isoforms at 24 hpi (**c**) and 48 hpi (**d**).

*2.5. Predicted the Protein Domains Alterations Caused by AS Occurred in #241-Specific DASGs*

Considering that AS may change the protein domain [15], we investigate the domain architecture of transcripts/isoforms produced by #241-specific DASGs. There were 68 #241-specific DASGs that underwent domain alterations due to AS, which corresponds to 312 transcripts/isoforms (Table S6). Among them, 56 #241-specific DASGs exhibited domain loss or gain. For example, three annotated transcripts (*Os06t0227200-01, Os06t0227200-02, and Os06t0227200-03*) of *Os06g0227200*, encode proteins with Dev_Cell_Death (PF10539) domains, which are absent in the novel isoform *MSTRG.20959.2*. A similar phenomenon was also detected in the bHLH TF gene (*Os01g0952800*) and the Glutathione S-transferase gene (*Os12g0210300*). Twelve #241-specific DASGs underwent domain changes, such as a gene encoding PPR repeat-containing protein (*Os02g0829850*). In addition, one gene encoding an EF-hand motif-containing protein, *Os02g0802400*, was found to have undergone domain gain.

*Os06g0227200* was predicted to encode a development/cell death-associated protein (Dev_Cell_Death: PF10539). Three annotated transcripts (*Os06t0227200-01, Os06t0227200-02, and Os06t0227200-03*) and one novel isoform (*MSTRG.20959.2*) were generated by *Os06g0227200*. The AS profile and expression profile of *Os06g0227200* revealed that one A3SS alternative splicing event occurred in the last exon of *Os06g0227200* (Chr6:6597193-6599496) at 24 hpi (Figure S2a; Table S3), which drove the primary transcripts/isoforms switching from *MSTRG.20959.2* to *Os06t0227200-03* (Figure S2c). PPI network analysis predicted that Os06g0227200 might interact with SKIPA, an SNW/SKI-interacting protein A that acts as a positive regulator of abiotic tolerance and a regulator of cell viability [27] (Figure S2b).

*2.6. Expression Analysis of Rice Spliceosome Involving in Regulation of AS Events in #241 during Interaction with M. oryzae*

In Eukaryotes, mature mRNAs are generated from precursor mRNAs (pre-mRNAs) through RNA splicing. This process is performed by the spliceosome, a highly dynamic ribonucleoprotein complex that contains small nuclear ribonucleo-proteins (snRNPs) and non-snRNP proteins [28]. To study spliceosomes involved in the regulation of AS, we retrieved annotated rice spliceosome protein sequences from the PlantGDB database (http://www.plantgdb.org/, accessed on 11 April 2022). A BLASTP search was then used to check for putative spliceosomes missing from the database. In total, 281 rice spliceosome coding genes were obtained (Table S7). Among them, 60, 117, 42, and 62 genes were assigned into subfamilies of the small nuclear ribonucleoprotein (snRNP) subfamily, the splicing factor subfamily, the splicing regulation subfamily, and the novel spliceosome proteins subfamily, respectively (Table S7). Thirteen and four spliceosome genes were significantly differentially expressed at 24 and 48 hpi (Figure S3a,b). Based on their expression analysis, eight spliceosome genes were significantly up-regulated at 24 hpi (Figure S3c). *Os02g0150100*, a gene that encodes one U4/U6, was up-regulated by 4.66-fold, followed by *Os03g0569900* and *Os02g0767100*, which encode one HSP73/HSPA8 and one CBP20, respectively. At 48 hpi, four up-regulated spliceosome genes were identified, which included *Os03g0110400* (encoding SF2/ASF protein) and *Os04g0504800* (encoding NPW38 protein) (Figure S3b).

## 3. Discussion

Alternative splicing (AS) occurs widely in eukaryotes and shapes the diverse proteome without gene expansion. At present, AS events are reported to be involved in multiple biological processes in rice, such as development and the responses to abiotic or biotic stresses [29–31]. However, it is still unclear about the genome-wide AS profile of rice in response to *M. oryzae*. In this study, we designed a transcriptome assembly pipeline to combine public RNA-seq data and explored the relationship between AS and durable rice resistance through the investigation and comparison of differentially alternatively spliced genes (DASGs) in *Pi21*-RNAi transgenic line (#241) and Nip.

Alternative donor (A3SS) splicing is the most common AS type in both the susceptible and resistible rice cultivars during interaction with *M. oryzae* Guy11, which is consistent with the results of previous research [32]. However, [26] found that intron retention (RI) was the most common AS type in *M. oryzae* during its infection procession. Thus, the different AS preferences of rice and *M. oryzae* were discovered in this study. Totals of 187 and 191 #241-specific DASGs were found at 24 and 48 hpi, which is more than were found in Nip. This suggests that AS may play an important role in durable rice resistance at an early interaction stage.

In detail, #241-specific DASGs at 24 hpi encoding protein kinase, leucine-rich repeat containing proteins, bHLH TFs and WRKY TFs, and ATPases were found involved in responses to plant hormone stimulation, such as jasmonic acid (JA) and ethylene (ET), which play an essential role in transducing the activation of plant defense systems [33]. We thus inferred that these DASGs might be related to the regulation of response to plant hormones.

According to [30], TF genes of WRKY and bHLH were also found to undergo frequent AS events in response to salt stress and pathogen infection. For example, [15] proposed that two WRKY TFs of the japonica rice cultivar "Zhonghua 17" (ZH17), *OsWRKY62* (*Os09g0417800*) and *OsWRKY76* (*Os02g0181300*), generate short alternative variants to enhance rice resistance against *M. oryzae*. However, we could not find *OsWRKY62* and *OsWRKY76* in the list of #241-specific DASGs, which may originate from different genetic backgrounds of the cultivars used. Similar regulation was also reported in the LAMMER kinase gene *OsDR11* (*Os12g0460800*), which produced two alternative variants, with contrasting functions, in response to *Xoo* infection [34]. We also found that this gene underwent an A3SS AS event and a retained intron in response to *M. oryzae* Guy11 infection (Table S2). Alternative splicing of *OsDR11* (*Os12g0460800*) may play an important role in rice defense against pathogens. Furthermore, *Os01g0952800*, encoding bHLH TF, generates three alternative variants with down-regulated expression levels, which may be involved in ethylene-activated signaling pathways. Notably, the wheat bHLH TF gene *TabHLH060* was also inhibited by ethylene and enhanced the susceptibility of transgenic *Arabidopsis thaliana* [35]. Thus, we suggested that *Os01g0952800* was inhibited by ethylene, which might be related to its alternative splicing events. Detailed annotation of *Os05g0425700* showed that a signature F-box motif at the N-terminus and leucine rich repeats (LRR_6) at the C-terminus, which represents that *Os05g0425700* encodes an F-box protein. In *Arabidopsis*, F-box protein Coronatine insensitive 1 (COI1) is a pivotal factor in the JA signal response [36]. However, how AS affects the F-box protein-regulating JA signal response is still unknown and this study provides insight into this aspect.

Among #241-specific DASGs at 48 hpi, *Os08g0295100*, a ubiquitin-like protein encoding gene, underwent an SE AS event in response to *M. oryza* infection and may be involved in response to endoplasmic reticulum (ER) stress. Three classes of ubiquitination pathway-associated enzymes were summarized by [37]. A BLAST search against the Uniprot database revealed that *Os08g0295100* encodes an E3 ubiquitin ligase. Under diverse stress, the demand for plant protein exceeds the plant system capacity, which results in the accumulation of misfolded or unfolded proteins and sets off ER stress [38]. In response to ER stress, unfolded protein response (UPR) was activated to lighten the load of unfolded proteins in ER, during which E3 ubiquitin ligase is required for marking unfolded proteins with adapters so that the ER-associated degradation (ERAD) system removes these proteins. In this study, we confirmed that multiple isoforms were generated by *Os08g0295100*, and primary isoforms switched from long-spliced variants (*MSTRG.25458.2*: 1020 bp) to short-spliced variants (*MSTRG.25458.1*: 630 bp), which may affect ubiquitination of unfolded target proteins and durable rice resistance. Furthermore, the PPI network of Os08g0295100 predicted that Os08g0295100 might interact with a SNARE protein (Uniprot ID: Q9LGF8). In *Arabidopsis thaliana*, SNARE protein SYP61 and ubiquitin ligase ATL31 cooperatively regulated the response to carbon/nitrogen conditions, during which SYP61 interacted with ATL31 and was ubiquitinated [39]. Thus, we suggested a similar cooperation with Os08g0295100 and Q9LGF8 may also play a role in response to rice blasts. However, the relationship between AS events of *Os08g0295100* and its function still needs further research.

Through investigation of infection-specific AS isoforms generated by #241-specific DASGs, the expression levels of 19 and 26 transcripts/isoforms were specifically dominant (TUI value > 0.5) at 24 and 48 hpi in the #241-*M. oryzae* interaction. *Os06t0227200-03*, the infection-specific primary transcript generated by *Os06g0227200* at 24 hpi, encodes protein with the Dev_Cell_Death (DCD) domain, which was originally detected in the soybean *NRP*-gene sequence and mediates signaling in plant development and programmed cell death (PCD) [40]. Interestingly, an A3SS AS event results in a primary transcript/isoform switch from *MSTRG.20959.2* to *Os06t0227200-03* during interactions with rice blasts (Figure S2a,c), which results in DCD domain gain (Table S6). Hence, it is reasonable to infer that *M. oryzae*-induced AS events enhance the *Os06t0227200-03* production, which activates PCD to limit the spread of *M. oryzae* invasive hyphae. Moreover, *Os07g0495900* encodes AAA-

type ATPase and underwent an A5SS AS event at 24 hpi, which resulted in primary transcript/isoform switching from the novel isoform *MSTRG.23603.1* to the annotated transcript *Os07t0495900-02*. Os07t0495900-02 was also associated with a response to JA (Figure 3a). The AAA-type ATPase from tobacco (*Nicotiana tabacum*), *NtAAA1*, was induced by JA and ethylene during the hypersensitive response (HR) resulting from infection by tobacco mosaic virus (TMV) and *Pseudomonas syringae* [41]. Thus, we suspect that JA may stimulate AS events, as shown by the induction of *Os07t0495900-02*.

Taken together, TFs of WRKY and bHLH, F-box protein with leucine rich repeats, and AAA-type ATPase underwent AS events and may be involved in the response to the stimulation of plant hormones, especially JA and ET. More importantly, we confirmed that skipped exon AS events in the E3 ubiquitin ligase coding gene *Os08g0295100* enhanced the generation of its short isoform *MSTRG.25458.1* during interaction with *M. oryzae*. *In silico* analysis revealed that Os08g0295100 may interact with the SNARE protein Q9LGF8, which might cooperatively regulate rice defense. This study will contribute to a greater understanding of the AS landscape in rice blast-susceptible and resistible *Oryza sativa*, which helps to explore AS events associated with durable rice resistance.

## 4. Method and Materials

### 4.1. Data Collection

Transcriptome data of *Pi21*-RNAi transgenic rice line (#241) and "Nipponbare" at 24 and 48 hpi of interacting with *M. oryzae* Guy11 were retrieved from the Ensembl website with accession of SRA492222, which includes millions of 50-bp single-ended RNA-seq reads. Detailed information of the collected data is provided in Table S1. *Oryza sativa* reference genome IRGSP-1.0 was downloaded from the Ensembl website (http://plants.ensembl.org/Oryza_sativa/Info/Index, accessed on 1 April 2022). FastQC v0.11.8 and Trimmomatic v.38 were used to assess read quality and to remove poor-quality reads or reads that consisted of adapters only.

### 4.2. Parse of Gffcompare Class Code to Distinguish Annotated Genes and Novel Genes/Isoforms

Clean DNA sequencing reads were mapped to the annotated rice genome using HiSAT2 with the parameters: no-mixed and no-discordant. The aligned reads for each sample were used to assemble the transcripts using StringTie v 2.1.1. The assemblies produced by StringTie were merged with the reference annotation file into one GTF file using the merge command in StringTie. The Transdecoder v5.5.0 was used to check the coding potential of the assembled transcripts among the StringTie-merged GTF files, and the assembled transcripts without protein-coding potential were filtered out. The remaining assembled transcripts were compared with the annotated transcripts for the classification of novel genes or novel transcripts using gffcompare v0.11.5, which was based on the gffcompare class code parse method proposed by [22].

### 4.3. Quantification of Transcript Abundances and Identification of Differentially Expressed Genes (DEGs)

The determination of the relative abundance of the annotated and assembled transcripts at different host–pathogen interaction stages was performed using StringTie, which outputs FPKM (fragments per kilobase of transcript per million mapped reads) values for each transcript. The expression profiles of each transcript were then converted into gene abundance using TBtools [42], with StringTie-merged GTF files as input. The differential gene expression analysis was performed using DEGseq. Genes with a combination of an adjusted $p$-value < 0.05 and $|$fold change$| \geq 1$ were considered to be differentially expressed.

### 4.4. Identification of Genes That Undergo AS Events

Among the AS detection tools available, SUPPA2 can give an effective and accurate prediction of AS events, especially at low sequencing depth and with short reads. In this study, we used SUPPA v2.3 to generate the AS profiles of genes in the StringTie-merged

GTF file, including annotated genes and novel genes based on the transcript assembly results. SUPPA first generates annotated files for seven types of AS events (A5SS, A3SS, SE, RI, MXE, AFE, and ALE). SUPPA2 then quantifies the percentage spliced index (PSI; $\psi$) for each sample, which indicates the inclusion of sequences into transcripts based on the normalized transcript abundance values (FPKM) and thus represents the AS event inclusion level. Transcripts with FPKM values < 1.0 were eliminated. The difference of splicing change ($\Delta$PSI; dpsi value) from each AS event across multiple interaction stages and their significance were calculated, which provided high-confidence AS events between the two conditions.

### 4.5. Gene Ontology Enrichment Analysis

The total proteins, encoded by the annotated and StringTie-merged transcripts, were uploaded on the eggNOG site (http://eggnog-mapper.embl.de/, accessed on 5 April 2022) and Gene Ontology (GO) annotation was performed. GO enrichment analyses were performed using TBtools. The GO terms with FDR < 0.05 and corresponding GO levels > 4 were retained and defined as enriched GO terms.

### 4.6. Visualization of the AS Profiles of the DASGs and Protein–Protein Interaction (PPI) Network

The gene models and corresponding Sashimi plots were visualized via the Integrative Genomics Viewer (IGV) (Robinson et al., 2017). Heatmaps were generated with the heatmap.2 function in R. The interacting proteins encoded by DASGs that were unique to the *Pi21*-RNAi transgenic rice line were identified by the STRING database (https://cn.string-db.org/, accessed on 9 April 2022) using the default parameters.

### 4.7. PCR Amplification

Conidia of *Magnaporthe oryzae* strain Guy11 were used for leaf inoculation of Tetep plants, another resistible indica rice cultivar [43]. Conidial suspensions were adjusted to 5 × 105 spores/mL, and 5 mls of the suspensions were sprayed onto the leaves of young plants. Uninoculated leaves of "Tetep" plants were used as the control samples, and leaves of "Tetep" plants at 24 and 48 h post-inoculatio (hpi) were used as the treatment samples. PCR amplification followed the method of [44]. Total RNA was isolated from rice leaves using the Qiagen RNAeasy Mini Kit (Qiagen Inc., Valencia, CA, USA) according to the manufacturer's instructions. The purified RNA was converted to cDNA by a Takara PrimeScriptTM RT reagent kit. PCR amplification primers were designed via primer version 4.0 (https://bioinfo.ut.ee/primer3-0.4.0/, accessed on 7 April 2022) and are also provided in Figure S1. PCR amplifications were performed on an Eppendorf G-storm GS2 Mastercycler. The amplification conditions were a 4 min denaturation step at 94 °C, followed by 32 cycles of 30 s at 94 °C, 30 s at 55 °C, and 1 min at 72 °C, with a final extension step of 10 min at 72 °C.

**Supplementary Materials:** The following supporting information can be downloaded at: https://www.mdpi.com/article/10.3390/agronomy12102414/s1, Table S1. Data accession and mapping rate of data used in this study; Table S2. #241-specific in Guy11_241_24h vs. 0h comparison; Table S3. #241-specific DASGs in Guy11_241_48h vs. 0 h comparison; Table S4. GO enrichment analysis for #241-specific DASGs might be associated with rice defense; Table S5. Annotation of primary transcripts/isoforms generated by #241-specific DASGs; Table S6. #241-specific DASGs underwent domain alterations; Table S7. Overview of rice splicesome used in this study; Figure S1. Multiple sequence alignment of transcript/isoform generated by *Os08g0295100*. P1 and P2 represent forward and reverse primers used for PCR amplification. L1 and L2 fragments represent exons absent in *MSTRG.25458.1*; Figure S2. Sashimi plot and protein–protein network analysis for Os06g0227200. (a) Sashimi diagram showing the exon-intron structure of representative annotated transcripts and novel isoforms. The red dashed box encloses the location of AS events. (b) The PPI network of the protein encoded by *Os06g0227200*. Green, red, and blue lines indicate interactions predicted from gene neighborhood, fusions, and co-occurrence. Light green, black, and dark blue lines indicate additional interactions inferred from text mining, co-expression, and protein homology. (c) Expression

profile of annotated transcript and novel isoforms produced by *Os06g0227200*; Figure S3. Comparison of rice spliceosome component genes that are significantly differentially expressed at (a) 24 hpi and (b) 48 hpi. Expression profile of rice spliceosome component genes specific to the (c) Guy11_241_24 h sample and (d) Guy11_241_48 h sample.

**Author Contributions:** Y.L. (Yongfeng Liu), D.L. and Z.Q., planned and designed the research; D.L. and Z.Q., performed the experiments; D.L., drafted this manuscript; Y.D., J.Y., M.Y., R.Z., H.C., X.P., T.S. and J.Q., participated in reviewing this manuscript; Y.L. (Youzhou Liu) and Z.Q., supervised the manuscript, whole research, and provided guidance. All authors have read and agreed to the published version of the manuscript.

**Funding:** This work was supported by funding to Y.L. (Yongfeng Liu), from the National Natural Science Foundation of China (Grant/Award Number: 31861143011). This work also received funding from The Revitalization Foundation of Seed Industry of Jiangsu [Grant/Award Number: JBGS(2021)005] and Jiangsu Modern Agricultural Technology System of Rice and Wheat Industry JAST [2021] 271.

**Data Availability Statement:** Not applicable.

**Conflicts of Interest:** The authors declare no conflict of interest.

## Abbreviations

| | |
|---|---|
| AS | Alternative splicing |
| TFs | Transcription factors |
| DASGs | Alternatively spliced genes |
| SE | Skipped exon |
| MX | Mutually exclusive exons |
| A5SS | Alternative 5′ splice-site |
| A3SS | Alternative 3′ splice-site |
| RI | Retained intron |
| AF | Alternative first exon |
| AL | Alternative last exon |

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
