# Peer review of "A Genome-Wide Alternative Splicing Landscape Specifically Associated with Durable Rice Blast Resistance"

_agronomy, doi:10.3390/agronomy12102414_

Round 1

Reviewer 1 Report

The manuscript by Liang et al. reports on the use of public transcriptome data for #241 and the Nip rice genotypes for the identification of AS events in a few transcription factor genes which could be associated with durable rice blast resistance. By comparing the AS landscape of #241 and Nip, they identified many differentially alternatively spliced genes (DASGs) specific to #241 at 24 and 48 hours of M. oryzae infection. Especially, the ubiquitin coding gene Os08g0295100 was confirmed to undergo AS in response to M. oryzae, in which the smallest transcript variant accumulates in response to infection.

The study is planned and executed well and overall presented nicely.

However, there are numerous mistakes in writing which need to be corrected. I have checked only the abstract for such mistakes. Authors must check the whole manuscript for such mistakes and correct them. For example:

1. Magnaporthe oryzae must be italicized throughout.

2. Name of genes, such as Pi21, should be italicized.

3. Line 18: "W" of "We" should be made small.

4. Line 19: abbreviation "TF" appears suddenly without being expanded previously.

5. Line 22: replace "were" at two places with "was". Accordingly, DASGs should be DASG, as it is a single gene.

6. Line 22-23: "predicted that involved in" should be "predicted to be involved in".

7. Line 27: "so that" should be "to".

8. Authors have used article "the" unnecessarily at several places. They should avoid doing it. For example, Line 97: Delete "the" from "The identification of rice differentially alternatively".

Major concerns:

1. The recent review by Ganie and Reddy 2021 (https://doi.org/10.3390/biology10040309) describes the AS events that have reported by 2021 in the interaction of rice-M.oryzae interaction. Have authors found any AS events in their study that are also reported in Ganie and Reddy 2021? Authors should Discuss it well in the Discussion portion.

2. In order to make their manuscript more impactful, authors might consider transforming a plant (e.g. Arabidopsis) with the transcript variant which becomes abundant during M.oryzae infection and check if that can confer resistance to the host.

Author Response

  1. Magnaporthe oryzae must be italicized throughout.

Reply: We are so sorry for this mistake and have changed all ‘Magnaporthe oryzae’ to italics.

  1. Name of genes, such as Pi21, should be italicized.

Reply: We are so sorry for this mistake and have changed all ‘Pi21’ to italics.

  1. Line 18: "W" of "We" should be made small.

Reply: We are so sorry for this mistake! and have adjusted this sentence and removed ‘we’ at line 18-19 of newest manuscript.

  1. Line 19: abbreviation "TF" appears suddenly without being expanded previously.

Reply: We are so sorry for this mistake and corrected it at line 19 of newest manuscript.

  1. Line 22: replace "were" at two places with "was". Accordingly, DASGs should be DASG, as it is a single gene.

Reply: We are so sorry for this mistake and corrected it at line 22-23 of newest manuscript.

  1. Line 22-23: "predicted that involved in" should be "predicted to be involved in".

Reply: We are so sorry for this mistake and corrected it at line 22 of newest manuscript.

  1. Line 27: "so that" should be "to".

Reply: We are so sorry for this mistake and changed “so that” to “to” at line 26.

  1. Authors have used article "the" unnecessarily at several places. They should avoid doing it. For example, Line 97: Delete "the" from "The identification of rice differentially alternatively".

Reply: Thanks for your advice! We have corrected this inappropriate usage of “the”

At line 154, 620 and 630 of newest manuscript.

Major concerns:

  1. The recent review by Ganie and Reddy 2021 (https://doi.org/10.3390/biology10040309) describes the AS events that have reported by 2021 in the interaction of rice-M.oryzae interaction. Have authors found any AS events in their study that are also reported in Ganie and Reddy 2021? Authors should Discuss it well in the Discussion portion.

Reply: Thanks for this advice! We have discussed AS events described by Ganie and Reddy 2021 at line 409-426. In summary, WRKY TFs, bHLH TFs and OsDR11 (Os12g0460800) were present in both their review and this article, which indicated that AS events of these genes may play an important role in rice defense against pathogens.

  1. In order to make their manuscript more impactful, authors might consider transforming a plant (e.g. Arabidopsis) with the transcript variant which becomes abundant during M.oryzae infection and check if that can confer resistance to the host.

Reply: Thanks for this advice! Yes, we will track how AS affect rice durable resistance by making transgenic line in our future researches. However, because we do not have the Pi21-RNAi (the transgenic line used in this study), we will try to ask for collaboration with the owner of Pi21-RNAi, which will help us to investigate how AS of candidates in this study (e.g. Os08g0295100) afftect Pi21-related rice resistance in our future researches.

Reviewer 2 Report

This research article reports the pattern of alternative splicing associated with the broad-spectrum disease resistance of a Pi21 knock down line (#241) to the blast fungus, M. oryzae. Through transcriptome analysis of a set of publicly announced RNA-seq data, the authors nicely characterized the genome-wide dynamics of the AS landscape, which will provide insight into the involvement of AS in rice disease resistance to blast.

Overall, this manuscript is well prepared with an excellent data analysis skill. I recommend publication of this manuscript in this journal if the authors will improve (or fix) the following minor defects of the current version.

1.       Grammatical error is frequently found throughout the main text. Please proofread carefully.   

2.       All the scientific names, such as M. oryzae, A. thaliana, etc., must be in italic.

3.       Pi21 is gene name, so must be italicized.

4.       Please address the parent of #241 in Introduction.

5.       Line 125: please provide the reference for ‘SUPPA2’.

6.       Line 134: please provide the full words of the abbreviations in Figure 1d at the end of the figure legend.

7.       Line 139: please double check ‘(139-397 DASGs)’ if it is right.

8.       Figure 2: please add letters ‘a’ and ‘b’ for each venn diagram.

9.       Line 163: should be ‘#241-specific’

10.   Figure 3: The genes in red boxes should be explained in the figure legend.

11.   Line 197: The reference for ‘STRING database’ should be cited.

12.   Line 219: ‘PFAM’ was not explained in earlier texts.

13.   Lines 231 – 240: I think this part should be merged to the next section (2.5.)

14.   Line 248: ‘the protein domains’”

15.   Lines 269 – 278: It is not clear if this paragraph describes #241 only. If yes, what is the pattern in Nip?

16.   Line 290: ‘resistant’ instead of ‘resistible’

17.   Lines 297 – 324: Most of the sentences in this part should be improved by fixing errors in grammar.  

18.   Lines 374 – 376: Is this part related to this study? For that datasets, different strains (248, 235, and 162), not Guy11, were used for inoculation.

19.   Lines 428 – 432: What is ‘Tetep’? Was this cultivar used in this study? If yes, you need to explain.

Author Response

. Grammatical error is frequently found throughout the main text. Please proofread carefully.

Reply: Thanks for this advice! We have checked grammatical errors throughout the main text carefully.

  1. All the scientific names, such as M. oryzae, A. thaliana, etc., must be in italic.

Reply: We are so sorry for this mistake and have changed scientific names and gene names throughout the newest manuscript.

  1. Pi21 is gene name, so must be italicized.

Reply: We are so sorry for this mistake and have changed all ‘Pi21’ to italics.

  1. Please address the parent of #241 in Introduction.

Reply: Thanks for this advice! In summary, #241 was transgenic line by silencing Pi21 gene, which exist in quantitative trait loci (we also added corresponding citation at line 112-114 and 779-780). Based on this, we added introduction of what is Pi21 gene (line 112-114); the explanation of why Pi21-RNAi showing enhanced resistance is at line 114-117; #241 was given ID/accession provided by owner of Pi21-RNAi transgenic line, which we explained at line 119-120.

  1. Line 125: please provide the reference for ‘SUPPA2’.

Reply: Thanks for this comment! we have provided reference for “SUPPA2” at line 182 and 800-801 of newest manuscript.

  1. Line 134: please provide the full words of the abbreviations in Figure 1d at the end of the figure legend.

Reply: Thanks for this advice! We have provided full words of the abbreviations in Figure 1d at line 202-204.

  1. Line 139: please double check ‘(139-397 DASGs)’ if it is right.

Reply: We are sorry for this mistake! We have checked the DASGs number in 2.2, and corrected this mistake in line 208 of newest manuscript. A total of 336 DASGs in Nip_Guy11_24 h_vs_0 h. A total of 329 DASGs in Nip_Guy11_48 h_vs_0 h.

  1. Figure 2: please add letters ‘a’ and ‘b’ for each venn diagram.

Reply: We are sorry for this mistake! And we have added letters ‘a’ and ‘b’ for venn diagram in Figure 2 of newest manuscript.

  1. Line 163: should be ‘#241-specific’

Reply: Thanks for this comment and we have corrected it at line 232 of newest manuscript.

  1. Figure 3: The genes in red boxes should be explained in the figure legend.

Reply: We are sorry for this mistake! and we have added explanation of red boxes and their color. You can find it in line 261-264 of newest manuscript.

  1. Line 197: The reference for ‘STRING database’ should be cited.

Reply: Thanks for this comment! we have provided reference for “STRING database” at line 279-280 and 807-809 of newest manuscript.

  1. Line 219: ‘PFAM’ was not explained in earlier texts.

Reply: Thanks for this advice! Because “PFAM” is the name of domain website, but not the abbreviations. We have clarified it in line 233-xx of newest manuscript.

  1. Lines 231–240: I think this part should be merged to the next section (2.5.)

Reply: Thanks for this advice! We have moved this part to section 2.5, which you can find it in line 302-303 of newest manuscript.

  1. Line 248: ‘the protein domains’”

Reply: Thanks for this advice! We have changed it in line 340 of newest manuscript.

  1. Lines 269-278: It is not clear if this paragraph describes #241 only. If yes, what is the pattern in Nip?

Reply: Thanks for this question! yes, this paragraph only describes #241. Because all candidates in this study focus on #241-specific DASGs, which means these genes underwent alternative splicing in #241 plant but not in Nip, and also differentially expressed in #241 plant. Based on the criteria, genes described at this part  do not undergo alternative splicing (Lines 341-352 of newest manuscript).

  1. Line 290: ‘resistant’ instead of ‘resistible’

Reply: We are sorry for this mistake! and we have changed all ‘resistant’ to ‘resistible’ in newest manuscript at line 226, 397 and 605.